# Impact of Neonatal Body (Dis)Proportionality Determined by the Cephalization Index (CI) on Gross Motor Development in Children with Down Syndrome: A Prospective Cohort Study

**DOI:** 10.3390/children10010013

**Published:** 2022-12-21

**Authors:** Asija Rota Čeprnja, Shelly Melissa Pranić, Martina Šunj, Tonći Kozina, Joško Božić, Slavica Kozina

**Affiliations:** 1Department of Physical and Rehabilitation Medicine, University Hospital Split, Spinčićeva 1, 21000 Split, Croatia; 2Department of Public Health, University of Split School of Medicine, Šoltanska 2, 21000 Split, Croatia; 3Department of Gynecology and Obstetrics, University Hospital Split, Spinčićeva 1, 21000 Split, Croatia; 4Department of Professional Studies, University of Split, Kopilica 5, 21000 Split, Croatia; 5Department of Pathophysiology, University of Split School of Medicine, Šoltanska 2, 21000 Split, Croatia; 6Department of Psychological Medicine, University of Split School of Medicine, Šoltanska 2, 21000 Split, Croatia

**Keywords:** longitudinal design, gross motor development, motor skills, body proportionality

## Abstract

Background: Children with Down syndrome (DS) lag behind typical children in the acquisition of developmental milestones, which could differ depending on body proportionality. We aimed to determine the difference in the acquisition of developmental milestones in children with DS with a disproportionate cephalization index (CI) compared to a proportionate CI. We created a motor development model that predicted milestone acquisition times. Methods: In this 20-year prospective cohort study, 47 children with DS aged 3 months to 5 years, followed up to 2020, were grouped according to the ratio of head circumference to birth weight (HC/BW) or CI into proportionate (CI < 1.1) and disproportionate (CI ≥ 1.1). We used a modified Munich Functional Developmental Diagnostic Scale that was assessed for reliability and content validity (Levene’s test and discriminant analysis) to determine 28 motor milestones. Linear regression was used to predict time to milestone acquisition, controlling for sex, maternal age, and birth weight. Results: Compared to proportionate CI, children with disproportionate CI were delayed in the milestone acquisition of a prone position by 2.81 months, standing before walking by 1.29 months, and a supine position by 1.61 months. Both groups required more time to reach standing after the acquisition of independent walking, but children with disproportionate CI reached those milestones later (4.50 vs. 4.09 months, *p* < 0.001). Conclusion: Children with disproportionate CI acquired milestones in a predictable order but slower than those with a proportionate CI. Our findings support the need to classify the degree of motor developmental delay in children with DS into unique functional groups rather than rely on clinicians’ arbitrary descriptions of the timing of developmental delays in children with DS.

## 1. Introduction

A child’s motor development is normally evaluated based on the time and age at which developmental milestones are reached [1]. Children with Down syndrome (DS) achieve all motor milestones, such as sitting, crawling, standing, and walking, in the same order as other children, but with a certain delay and greater variability in time range compared to typical children [2]. Children with DS sit independently at an average age of 9 months (range: 6–16 months), stand independently at 18 months (12–38 months), and walk independently at the age of 24 months (16–42 months) [3]. 

Two theories have been posited to explain the delay in motor development among children with DS. The first is the intrapopulation theory, where differences in the acquisition of motor milestones in children with DS are attributed to early health issues and various surgeries for associated medical problems, which can have a negative impact on motor development [4]. Moreover, long hospital stays can limit a child’s exposure to external stimuli and inhibit brain development [5]. The second theory explains interpopulational differences between typical children and children with DS, where perhaps low muscle tone and the hypermobility of the joints of children with DS are at fault [2]. Further, previous studies showed that structural and functional disorders of the central nervous system can delay motor development [6,7]. For example, these disorders include changes in the shape and number of neurons, changes in the cerebrum size, hypoplasia and reduced granule cell density of the cerebellum, delayed myelination, and the pathophysiological processes caused by excessive gene expression at chromosome 21 [6,7].

Additionally, Nishizava et al. hypothesized that motor delay is caused by the slow acquisition of antigravity activities in children and applied a photo-elastic method to quantitatively evaluate antigravity functions in typical children and children with DS [8]. Based on contact pressure with the head, body, and upper and lower limbs in supine and prone positions, they concluded that children with DS have different antigravity functions compared to typical children. In prone positions, the head weight load decreases with age in healthy children, while in neonates with DS, where the reduction is significantly lower, the children had difficulties lifting their heads off a surface [8]. Head control is a fundamental motor skill in early development [9]. The aforementioned antigravity movements generally develop first in the head and are followed by development in the trunk (cervical to thoracic to lumbar) and then in the lower extremities. Head movement initiates antigravity activity in the supine or prone positions [10]. 

Some researchers believe that delayed motor development in typical children is caused by small head circumference (HC) [11] or low birth weight (BW) [12], as well as the disproportionate ratio between head circumference and body weight of neonates [13]. However, recent research suggests that HC is not associated with gross motor function [14]. Nishi et al. [15] used a proportionality index based on the ratio of head circumference to birth weight (HC^3^/BW) and asserted the use of this calculation to assess head size to weight in infants. The index was reliable in the average proportionality index through assessments from birth to 18 months of age, regardless of sex and race. Meanwhile, Harel [16] and Simic [17] similarly calculated neonatal proportionality by looking at the HC and BW and further described the relationship as the cephalization index (CI). They also highlighted the predictive validity of the CI for an increased risk of severe psychomotor impairment at age 3 years and during primary school [18,19]. Neonate disproportionality (CI ≥ 1.1) can delay a child’s motor development. Head weight load in prone and supine positions in disproportionate children can lead to issues with delay in establishing head control, a fundamental motor task of early development and one of the first antigravity activities acquired by children.

Despite the potential clinical usefulness of the CI, no studies exist that determine whether intrapopulational differences in acquiring motor development milestones are influenced by differences in head circumference to body weight ratio (CI < 1.1 or CI ≥ 1.1) in children with DS. Prior investigations in different countries focused on typical children and did not assess their CI [11,12]. It is unknown whether the CI of children with DS predicts future gross motor development. Thus, our study intended to address this gap by seeking to determine whether there is a difference in the acquisition of developmental milestones in children with DS who have a disproportionate CI compared to those with a proportionate CI. Additionally, we sought to produce a model for the prediction of the development of gross motor skills specific to children with DS according to their neonatal (dis)proportionality (CI ≥ 1.1). 

## 2. Materials and Methods

### 2.1. Participants 

The Ethics Committee at the University Hospital of Split approved this study. We obtained written informed consent from parents before the study. This study was carried out according to the principles established by the Declaration of Helsinki and was registered on ClinicalTrials.gov NCT03553706.

In this prospective longitudinal cohort study, the age of the acquisition of motor development milestones was monitored in 56 children with DS, aged from 3 months to 5 years at the time of enrollment or baseline, and followed up between 2000 and 2020. The sample size was not predetermined; rather, the children were selected by convenience. The research was conducted in the Department of Physical and Rehabilitation Medicine at the University Hospital Split in Croatia, where parents were invited to have their children, who were routine patients, participate in the study. Data were extracted during regularly scheduled neurodevelopmental therapy sessions by experienced clinicians. Comorbid conditions were assessed based on diagnoses made immediately after birth at the neonatology unit or from medical records. We excluded children with DS with major comorbid conditions according to the European Concerted Action on Congenital Anomalies and Twins (EUROCAT) classification [20] to reduce potential confounding in the relationship between body proportionality and developmental milestones. The comorbid conditions were a ventricular septal defect, endocardial cushion defects, a secundum atrial septal defect, anal atresia, duodenal atresia, hydrocephalus, and tetralogy of Fallot. Infants who missed more than three consecutive evaluations or all therapy sessions by the time of a milestone or group of milestones were excluded from the study. Clinical examinations prior to walking were carried out at 2- to 3-month intervals, and every 3–6 months after walking. The age of a child was calculated at every examination and expressed in days and months. We also recorded the acquisition times of specific motor milestones. 

### 2.2. Anthropometry

Trained nurses used standardized protocols to collect direct anthropometric parameters, including birth weight (BW), length, and head circumference (HC) for gestational age and sex of the participants. We used WHO Growth Charts to express anthropometric measurements for the children into age- and sex-specific percentiles. For premature children born < 37 weeks, we used Fenton’s growth curve [21] to adjust for sex and gestational age. Two authors (ARC and MŠ) additionally classified the children according to the CI defined as HCx100/BW using the cutoff points previously defined by Harel [16] and Simic [17].

### 2.3. Study Instruments and Measurements

We used a modified electronic version of the Munich Functional Developmental Diagnostics (MFDD) [22] to evaluate motor development. We chose the MFDD due to the comprehensiveness of the milestones that can be assessed as opposed to other existing instruments [23,24]. Table 1 shows the motor skills that we assessed in the children with DS according to different developmental periods. 

The HC and BW were obtained separately from clinical examinations by specially trained staff using calibrated instruments according to standardized protocols once a week at the first and subsequent visits. Based on the CI, infants were divided into groups that were either disproportionate (CI ≥ 1.1) or proportionate (CI < 1.1) at birth. 

The reliability and content validity of the MFDD for the evaluation of motor development in participants who had proportionate and disproportionate CI were assessed. We used Levene’s test of homogeneity to determine the discrimination of MFDD motor scales in determining motor milestone acquisition time (28 milestones) in children with proportionate and disproportionate CIs. Discriminant analysis was used to find a linear combination of features that characterizes newborns with DS or separates two or more classes of newborns with DS who had a CI < 1.1 or CI ≥ 1.1.

### 2.4. Outcomes

The primary outcome was the time to attainment of motor milestones in children with DS according to their CI established during routine examination. The secondary outcome was the anthropometric parameters including BW, length, and HC for gestational age and sex of the children with DS.

### 2.5. Statistical Analysis

We used binomial logistic regression to determine whether the acquisition of motor development milestones differed between the two groups. We reported odds ratios (OR) along with 95% confidence intervals. The outcome variable was the proportionality of the children, and the predictor variables included the scores from the MFDD. The MFDD scores for the 28 developmental milestones were analyzed as continuous variables. We controlled for maternal characteristics and children’s birth weight and sex. We reported frequencies, means ± standard deviation, medians with the interquartile (IQR) range, as well as 95% confidence intervals (CI). We determined the reliability and homogeneity of the variance of the results with Levene’s test and discriminant analysis. 

Linear regression was used to predict the number of months needed to reach 28 milestones in prone (“crawling age”), standing (“walking age A and B”), supine, and sitting (“sitting age”) positions for both children with proportionate and disproportionate CIs. Our regression model (*y* = b + m*x*) attempted to predict the expected months to reach a milestone from the obtained milestone scores, where *y* as the outcome variable is expected months for a function, *b* is the intercept, and *m* is the slope. The outcome and all explanatory variables were treated as continuous variables. Our model was good at predicting months to reach a milestone based on the residual distribution of months to each milestone vs. milestone scores (*p*-range 0.001–0.011). There was a linear relationship among the developmental milestones. We used R (version 4.0.2) for statistical analysis and considered *p*-value < 0.05 as significant.

## 3. Results

### 3.1. Participant Characteristics

Out of the 56 enrolled children, we excluded 9 (16%) children due to the misclassification of the CIs according to the discriminant analysis (Appendix A). Table 2 shows the characteristics of the 47 children included in the analysis for the current study. 

The average number of examinations per child was 11.14 ± 3.98 (624 examinations for 56 children), of which 8.02 ± 2.53 (449 examinations for 56 children) were performed before reaching the walking milestone and 3.13 ± 2.86 (175 examinations for 56 children) thereafter. The average number of physical therapy procedures per child was 166.64 ± 106.99 (9332 procedures for 56 children), of these 105.98 ± 75.68 (5935 procedures for 56) were performed before walking, and 60.66 ± 77.53 (3397 procedures for 56) thereafter. 

### 3.2. Acquisition of Motor Development Milestones in Disproportionate Compared to Proportionate CIs

Table 3 shows that children with a proportionate CI who acquired early motor milestones were more likely to acquire the motor skills involving head lifting 90° with forearm rest, extended arm support, and semiflexion of the hips and knees earlier than those with a disproportionate CI.

Concerning later motor milestones, children with a proportionate CI acquired all milestones, except standing on one foot without help for 2 s, earlier than children with a disproportionate CI (Table 4).

Figure 1 shows the months needed to acquire various milestones in the prone position, standing, and in the supine position in children with DS who have a proportionate CI and a disproportionate CI. 

### 3.3. Validity and Reliability of the Modified MFDD

The modified MFDD had good psychometric properties, as indicated by its high reliability (Cronbach’s alpha = 0.933) and content validity (0.930–0.931) (Appendix A). The content validity describes how closely the modified MFDD represents the original scale in assessing milestones. Specifically, the correlations include the individual milestones and the time of acquisition of walking independently (Appendix A). The homogeneity of the variance of the time to achieve 28 milestones in children with a proportionate CI and a disproportionate CI showed that the time to acquire 27 milestones did not deviate in the two groups, making them homogeneous (*p* > 0.05), apart from crawling/reciprocal creeping (*p* < 0.05) (Appendix A). This suggests that there was good discriminant validity of the modified MFDD scales. 

### 3.4. Comparison of the Discriminant Analysis and Clinicians’ Assessments 

After conducting the discriminant analysis, the division of the 56 children into two groups based on the proportionality of their anthropometric measures was confirmed against clinicians’ assessments of the CI (Wilks’ λ = 0.46; χ2 [df = 28] = 30.640, *p* = 0.333) (Appendix A): 86.7% (24/27) of them were properly classified in the proportionate group, and 13.3% (4/30) were wrongly classified (see Appendix A). In the disproportionate group, 80.8% (21/26) were correctly classified, and 19.2% (5/26) were wrongly so.

### 3.5. Milestone Characteristics of the Infants

We assessed 47 children classified as having a disproportionate CI (n = 21) or proportionate CI (n = 26) for milestone development. The average acquisition time of motor development for almost all milestones according to the modified MFDD was greater for disproportionate CIs than proportionate CIs (*p* < 0.05), as shown in Table 3, Table 4 and Table 5. 

We used the acquisition time of an earlier milestone to explain the variation in the mean months of a later milestone. Notably, disproportionate children were more delayed in “crawling age”, “walking age” A and B, and “sitting age” than proportionate children (Table 3, Table 4 and Table 5). Compared to children with a proportionate CI, those with a disproportionate CI had a 2.7-month delay in prone postures (head lifting 90° with forearm rest), a 2.4-month delay in supine postures (following a toy with eyes), and a 1·6-month delay in standing (semiflexion of the hips and knees) (Figure 1). Children with a disproportionate CI acquired the prone position at 2.68 months (*p* = 0.075), accounting for 71% of the variance around the mean number of months to that milestone. Children with a proportionate CI acquired the prone position at 3.25 months (*p* = 0.013), accounting for 90% of the variance around the mean months to that milestone (Figure 1). Regarding the supine position, children with a disproportionate CI acquired this milestone at 2.26 months (*p* = 0.011), while those with a proportionate CI did so at 2.39 months (*p* = 0.009). In each of these models, the dis/proportionality of the CI explained 84% and 85%, respectively, of the variation in the mean months to the supine position. For standing posture A, 4.56 months was the time that children with a disproportionate CI acquired this milestone (*p* = 0.001), whereas the time was 4.60 months for those children with a proportionate CI (*p* = 0.002) (Figure 1). We found 94% and 93% variation in the mean time, explained by the standing posture A in disproportionate CI and proportionate CI. Children with a disproportionate CI acquired the standing posture B later than those with a proportionate CI (4.50 and 4.09 months, *p* < 0.001 for both), where the standing posture B explained 91% of the variation in the mean months for disproportionate CI. Standing posture B was the variable that explained 91% of the variation in the mean months for children with a proportionate CI (Figure 1).

## 4. Discussion

Our model of motor development in children with DS provides a normative framework for predicting motor milestones in this population. With the knowledge that all children with DS are developmentally delayed, not just children with disproportionate CIs, the ability to predict the time to prone, supine, and standing positions would enable clinicians to better track motor development in children aged 3 months to 5 years with DS who have proportionate and disproportionate CIs. Our model of motor development by the proportionality of the CI allows clinicians to identify when an unexpected delay occurs, elucidating whether delays are more behind than expected compared to other children with DS. Accordingly, adequate early intervention could be developed to help children with DS through timely physical therapy.

It was observed in this study that, compared to proportionate children, disproportionate children had a 2.8-month delay in reaching the milestones of prone postures (head lifting 90° with forearm rest), a 1.6-month delay in supine postures (following a toy with eyes), a 1.3-month delay in standing (semiflexion of the hips and knees), and a 2.3-month delay in standing (walking independently). Additionally, there were significant developmental delays between children with proportionate and disproportionate CIs in the acquisition of extended arm support, semiflexion of hips and knees, and the ability to follow a toy with their eyes. Head control in the prone position is the earliest antigravity control that children have to develop [8]. A child raises their head by using cervical extensors. The power also depends on the cervical flexors, anterior muscles, to lengthen through reciprocal inhibition. Even with stronger cervical extensors, a child with DS is unable to lift his or her head without stabilizing it with another part of the body, which accounts for the delay in the first milestone and the supine position [25]. 

One study to date has created a growth curve that predicts the motor development of children with DS [26]. Lauteslager and colleagues found a 50% probability that children with DS would achieve sitting at age 22 months and crawling at 25 months with the use of a validated growth curve based on the Test of Basic Motor Skills for Children with Down Syndrome (BMS) scale, a scale that assesses 15 motor development skills in children with DS [26]. Although the MFDD scale was not validated for children with DS at this study’s initiation, it was chosen for its comprehensiveness of developmental milestones. Additionally, in contrast to the Lauteslager [26] study, we compared the time to reach motor milestones between children with DS who have disproportionate and proportionate CIs, whereby the proportionality of the children could explain differences in motor development.

Generally, the children with DS in our study found it more difficult to achieve motor milestones in the prone position than in the supine position. These results may be aligned with previous investigations about the delay in achieving motor milestones in children with DS, although these studies used different scales and scoring to assess milestones than what was used in the present study [26,27]. Prone positions require greater muscle antigravity power [8]. Prior research has shown that the order in which the prone posture is acquired is different in children with DS compared to typical children [28]. Tudella and colleagues [28] found that children with DS acquired the prone position in their 8th and 9th months, compared to typical children, who usually acquire the prone position in the 6th month. For instance, children with DS in our study acquired “Rolls both ways” before they acquired “Extended arm support” and “Sits alone stably” before “Sits down alone”. Our findings support interpopulation interpretations from previous studies that suggested that CI physical conditions are more instrumental in the delay in motor milestone acquisition than chronic diseases or CNS irregularities [2,4,6,7]. 

The prevailing opinion in the literature is that children with DS acquire milestones in the same order as typical children [28,29]. In our study, children with DS adopted rotations before leaning on extended arms. Leaning on outstretched arms proved to be a more demanding task because it required greater antigravity activity, which was insufficient in the examined children. Of the noted difficulties regarding acquiring motor milestones, children with DS experience difficulties contracting the flexors of the neck, trunk, and upper and lower limbs, particularly at one year of age. They stay in static and relaxed positions for extended periods and fail to activate antigravity muscles in prone and supine positions [8,27,30]. Despite these difficulties, children with Down syndrome acquire the ability to rotate with trunk dissociation at an average age of 8 months, as do typical neonates [31]. Antigravity muscles (the erector spinae, gluteus maximus, proximal hamstrings, and quadriceps muscles) keep a person upright against gravity, whether sitting, kneeling, quadruped, or standing erect. In the supine position, the flexors act as antigravity muscles (cervical flexors, abdominal muscles, and hip flexors) [9]. Children who had disproportionate CIs walked independently at the same time as proportionate children, but afterward experienced delays in milestone acquisition in standing posture B. The most complex and difficult motor function to evaluate was jumping in place, which develops last in children with DS, like in typical children, except with delays [28]. In the acquisition thereof, children with a disproportionate CI were 6.63 times more likely not to adopt milestones compared to proportionate children. 

Similar to children with DS, children with an overall developmental delay, cerebral palsy, or who were born prematurely also experience delays in motor development [32]. Interventions to improve milestone development in children with motor delays exist for cerebral palsy, but their appropriateness and effectiveness remain unknown. Due to the absence of a standardized method to time developmental delays in children with DS thus far, clinicians provide interventional therapy when a delay is noticed in routine care. Interventions to improve motor developmental delay in children with DS should focus on the delivery of appropriate and timely physical therapy [33].

### Limitations

Despite the prospective study design that allowed the collection of potentially confounding factors, we note that there are limitations to the current study. We did not determine the interrater reliability between the assessments of the clinical staff and rater bias during the evaluation of the weight and head circumference of the children, which could have played a role in shifting the weights in a particular direction. However, clinicians involved in the data collection followed standard protocols for the procedures, which likely minimized differences between the raters. In estimating the degree of delays, we did not use terms such as short- or long-term or permanent as qualitative descriptors but described the delays quantitatively because we did not have a reference framework for the acquisition of gross motor milestones specifically for other forms of disproportionality (e.g., small for gestational age or large for gestational age). Moreover, the wide range in the months to the acquisition of milestones did not indicate the degree of severity of the delay. Although we excluded children with comorbid conditions from the current study, there could be other unaccounted factors that could confound the relationship between body proportionality and developmental milestones. Furthermore, there may be additional indicators of developmental delay that the MFDD did not capture but were distinguishable to clinicians. 

We used the CI (HC/BW) as a measure of dis/proportionality. Nine children in the current study were misclassified according to discriminant analysis (Appendix A). The misclassified group had heterogeneous anthropometric measures and displayed other dis/proportionalities (BW/BL, body length) and ponderal index (PI)), including those according to the CI.

Future research could examine (a) whether acquired motor milestones are dependent on other forms of body disproportionality and (b) if such specificities of motor milestone development also exist in typical children with regard to the type of disproportionality. Additionally, a future study could determine how differences between children with DS with a proportionate CI and a disproportionate CI persist over time or affect development in other domains.

## 5. Conclusions

The determination of the time when a delay could occur would be beneficial in clinical practice, and the motor development model described in the current report may provide clinicians with a way to deliver appropriate and timely interventions that are not decided arbitrarily. Future studies could utilize the motor development model to determine when to create precise interventions for disproportionate and proportionate children with DS toward the attainment of the key motor milestones.

## Figures and Tables

**Figure 1 children-10-00013-f001:**
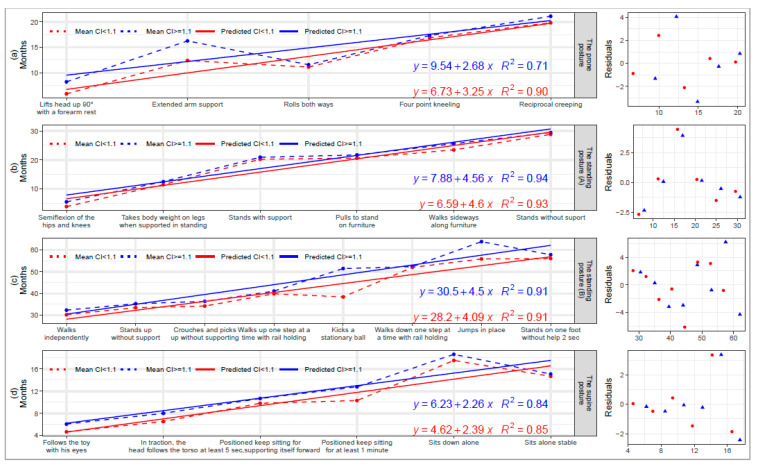
Linear dependency in acquisition of motor development milestones in prone position (crawling age), standing posture (walking age A and walking age B), supine position, and sitting (sitting age) for proportionate and disproportionate children with DS. (**a**) Mean months to reach milestones in prone position according to CI. (**b**) Mean months to reach milestones while in standing A according to CI. (**c**) Mean months to reach milestones while in standing B according to CI. (**d**) Mean months to reach milestones while in supine position according to cephalization index (CI). “A” denotes elementary motor skills that prepare a child for more advanced motor skills. “B” denotes when children start walking and performing more advanced motor skills. Abbreviations: CI, cephalization index (CI).

**Table 1 children-10-00013-t001:** General motor skills of children according to the Munich Functional Developmental Diagnostics (MFDD) ^1^.

**Posture**	**Developmental Period**	**Motor Skills**
Prone	Crawling age	Child can lift their head 90° while resting on their forearms, pushes their chest up with outstretched arms (extended arm support)
		Rises to their knees and palms (four-point kneeling)
		Rolls over on the abdomen and vice versa (rolls in both directions)
		Crawls on hands and knees (reciprocal creeping)
Standing	Walking age, A ^2^	Semiflexion of the hips and knees
		Child’s legs can support their body weight while standing with support
		Stands with or without support
		Pulls themselves to stand up with the support of furniture
		Walks along sideways with the support of furniture
Supine and sitting	Walking age, B ^3^	Walks independently (walks alone)
		Stands up without support
		Crouches (bends) and picks something up without support
		Walks up or down one step at a time while holding onto a railing
		Kicks a stationary ball
		Stands on one foot without help for 2 s and jumps in place
	Sitting age	Child follows a toy with their eyes
		The head follows the torso in traction
		Child can sit for at least five seconds
		Child can support themselves leaning forward
		Sits down stably alone

^1^ Hellbrügge T. [22]. ^2^ Walking age A denotes elementary motor skills that prepare a child for more advanced motor skills that are a part of walking age B. ^3^ Walking age B denotes when children start walking and performing more advanced motor skills.

**Table 2 children-10-00013-t002:** Characteristics of proportionate (cephalization index (CI) < 1.1) and disproportionate (CI ≥ 1.1) children with Down syndrome.

Sociodemographic and Birth Characteristics	Groups of Children	
	CI < 1.1 (n = 26)	CI ≥ 1.1 (n = 21)
Sex, n (%)		
Female	10 (38.46%)	10 (47.62%)
Male	16 (61.54%)	11 (52.38%)
Gestational age (weeks), mean (SD) ^1^	38.31 (1.12)	36.57 (1.88)
Postnatal anthropometric measures, mean (SD)		
Body weight (g)	3363.46 (326.81)	2365.48 (447.74)
Body length (cm)	49.73 (1.59)	45.10 (2.79)
Head circumference (cm)	33.31 (1.60)	31.46 (1.71)

^1^ SD—standard deviation.

**Table 3 children-10-00013-t003:** Comparison of early motor milestone acquisition between children with DS with proportionate and disproportionate CIs.

Motor Skills in the Prone Posture “Crawling Age”								
Milestones	A(Symmetry)	N	Months (Mean)	SD ^1^	OR ^2^ (CI 95%)	χ² Tests	df	*p*-Value
Lifts head up 90° with forearm rest	CI ^3^ < 1.1	26	5.85	2.58				
	CI ≥ 1.1	21	8.22	3.22	4.56 (2.83–7.32)	41.30	1	<0.001
	Total	47	6.91	3.09				
Extended arm support	CI < 1.1	26	12.40	3.45				
	CI ≥ 1.1	21	16.28	4.59	7.34 (5.10–10.6)	128.00	1	<0.001
	Total	47	14.13	4.41				
Rolls both ways	CI < 1.1	26	11.13	3.37				
	CI ≥ 1.1	21	11.58	3.81	1.38 (0.977–1.94)	3.34	1	0.068
	Total	47	11.33	3.54				
Four-point kneeling	CI < 1.1	26	16.90	4.61				
	CI ≥ 1.1	21	17.32	5.44	0.986 (0.745–1.31)	0.01	1	0.923
	Total	47	17.09	4.95				
Reciprocal creeping	CI < 1.1	26	19.84	3.80				
	CI ≥ 1.1	21	21.12	6.66	1.93 (1.48–2.51)	23.60	1	<0.001
	Total	47	20.41	5.25				
**Motor Skills in the Standing Posture “Walking Age” A**								
Semiflexion of the hips and knees	CI < 1.1	26	3.90	2.51				
	CI ≥ 1.1	21	5.53	3.68	3.51 (1.71–7.21)	12.60	1	<0.001
	Total	47	4.63	3.16				
Holds body weight on legs when supported in standing	CI < 1.1	25	11.47	4.47				
	CI ≥ 1.1	20	12.50	3.75	2.27 (1.58–3.27)	19.80	1	<0.001
	Total	45	11.93	4.15				
Stands with support	CI < 1.1	24	20.22	5.48				
	CI ≥ 1.1	19	20.91	5.02	2.97 (2.23–3.94)	58.10	1	<0.001
	Total	43	20.53	5.23				
Pulls to stand on furniture	CI < 1.1	24	20.62	5.63				
	CI ≥ 1.1	19	21.70	6.46	2.37 (1.80–3.11)	39.00	1	<0.001
	Total	43	21.10	5.96				
Walks sideways along furniture	CI < 1.1	22	23.45	3.87				
	CI ≥ 1.1	18	25.59	6.83	3.14 (2.38–4.15)	67.90	1	<0.001
	Total	40	24.41	5.44				
Stands without support	CI < 1.1	23	28.81	5.81				
	CI ≥ 1.1	18	29.44	7.27	0.972 (0.769–1.23)	0.06	1	0.810
	Total	41	29.09	6.41				

^1^ SD—standard deviation. ^2^ OR—odds ratio. ^3^ CI—cephalization index.

**Table 4 children-10-00013-t004:** Comparison of later motor milestones in children with DS who have a proportionate CI compared to a disproportionate CI.

Motor Skills in the Standing Posture “Walking Age” B								
Milestones	A(Symmetry)	N	Months (Mean)	SD ^1^	OR ^2^ (CI 95%)	χ² Tests	df	*p*-Value
Walks independently	CI ^3^ < 1.1	23	30.23	5.96				
	CI ≥ 1.1	17	32.34	6.78	2.04 (1.60–2.59)	34.50	1	<0.001
	Total	40	31.12	6.32				
Stands up without support	CI < 1.1	22	33.47	6.75				
	CI ≥ 1.1	17	35.29	9.84	0.518 (0.416–0.645)	34.90	1	<0.001
	Total	39	34.26	8.17				
Crouches and picks something up without supporting	CI < 1.1	23	34.23	11.60				
	CI ≥ 1.1	17	36.35	7.82	2.15 (1.73–2.67)	47.70	1	<0.001
	Total	40	35.13	10.10				
Walks up one step at a time with rail holding	CI < 1.1	20	39.84	11.98				
	CI ≥ 1.1	14	41.06	7.87	2.02 (1.63–2.52)	40.70	1	<0.001
	Total	34	40.34	10.36				
Kicks a stationary ball	CI < 1.1	18	38.41	8.65				
	CI ≥ 1.1	12	51.42	16.14	7.41 (5.50–9.98)	203.00	1	<0.001
	Total	30	43.61	13.59				
Walks down one step at a time with rail holding	CI < 1.1	17	51.93	14.79				
	CI ≥ 1.1	14	52.28	12.70	1.27 (1.04–1.55)	5.73	1	0.017
	Total	31	52.09	13.66				
Jumps in place	CI < 1.1	10	55.81	9.55				
	CI ≥ 1.1	6	63.74	17.21	6.63 (4.58–9.61)	117.00	1	<0.001
	Total	16	58.79	13.01				
Stands on one foot without help for 2 s	CI < 1.1	11	56.00	10.87				
	CI ≥ 1.1	6	57.74	11.57	0.834 (0.638–1.09)	1.76	1	0.184
	Total	17	56.61	10.79				

^1^ SD—standard deviatio*n.*
^2^ OR—odds ratio. ^3^ CI—cephalization index.

**Table 5 children-10-00013-t005:** Comparison of the acquisition of sitting and protective extension reflex motor skills in children with DS with a proportionate CI and a disproportionate CI.

Motor Skills in the Supine and Sitting Posture “Sitting Age”								
Milestones	A(Symmetry)	N	Months (Mean)	SD ^1^	OR ^2^ (CI 95%)	χ² Tests	df	*p*-Value
Follows a toy with eyes	CI ^3^ < 1.1	26	4.66	2.65				
	CI ≥ 1.1	21	6.07	3.17	1.96 (1.14–3.35)	6.01	1	0.014
	Total	47	5.29	2.95				
In traction, the head follows the torso	CI < 1.1	26	6.53	2.46				
	CI ≥ 1.1	21	8.00	2.52	2.26 (1.37–3.74)	10.40	1	0.001
	Total	47	7.19	2.57				
Positioned, keeps sitting for at least 5 s supporting self forward	CI < 1.1	26	9.82	9.19				
	CI ≥ 1.1	21	10.68	3.81	4.68 (3.18–6.88)	64.60	1	<0.001
	Total	47	10.21	7.24				
Positioned, keeps sitting for at least 1 min	CI < 1.1	26	10.31	2.10				
	CI ≥ 1.1	21	12.78	3.91	2.51 (1.71–3.70)	22.40	1	<0.001
	Total	47	11.41	3.25				
Sits down alone	CI < 1.1	25	17.51	3.41				
	CI ≥ 1.1	21	18.62	4.10	1.80 (1.36–2.39)	17.10	1	<0.001
	Total	46	18.02	3.74				
Sits alone stably	CI < 1.1	26	14.69	3.11				
	CI ≥ 1.1	21	15.09	3.91	1.20 (0.885–1.61)	1.35	1	0.245
	Total	47	14.87	3.46				
**Protective extension reflex**								
Forward	CI < 1.1	26	10.77	2.46				
	CI ≥ 1.1	21	13.60	5.23	5.99 (3.99–8.99)	82.30	1	<0.001
	Total	47	12.03	4.15				
Sideways	CI < 1.1	26	15.50	3.18				
	CI ≥ 1.1	21	18.03	6.27	2.86 (2.10–3.90)	45.80	1	<0.001
	Total	47	16.63	4.92				
Backward	CI < 1.1	20	27.54	6.20				
	CI ≥ 1.1	16	24.94	6.48	0.810 (0.618–1.06)	2.36	1	0.124
	Total	36	26.38	6.37				

^1^ SD—standard deviation. ^2^ OR—odds ratio. ^3^ CI—cephalization index.

## Data Availability

The data used in this study are not publicly available. All requests for data must be justified and will be reviewed by the authors. The participant-level data used in this study can be requested from Rota Čeprnja with a protocol clearly describing the methodology. Researchers will be required to sign a data access agreement.

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
