# Peer review of "Impact of Neonatal Body (Dis)Proportionality Determined by the Cephalization Index (CI) on Gross Motor Development in Children with Down Syndrome: A Prospective Cohort Study"

_children, 2022, doi:10.3390/children10010013_

Round 1

Reviewer 1 Report

This is a robust longitudinal study that follows the development of 56 children with Down syndrome (DS) over 20 years. That is the strength of the overall study but it is not fully described in this paper as the authors only focus on early motor milestones and do not examine how differences between the proportional and disproportional CI children persist over time or affect long-term development (e.g., speech and language, intelligence, social skills, adaptive behaviors). The paper would be much stronger if they also demonstrate how greater delays in early motor development affect other cognitive domains or persist into middle- or late-childhood or adolescence.

There are two aims of the study. One, to compare the motor development of children with DS who have proportional head circumference to body mass and those who do not. Two, to test the psychometric properties of the MFDD, which has not been validated for children with DS. The design of the study is able to answer these two aims.

There's inconsistent usage of terminology in the abstract. At times, the authors use disproportional/proportional, other times, disproportionate/proportionate. These terms by themselves are somewhat vague as they are adjectives, not nouns. I recommend the authors include "CI" (i.e., disproportionate CI) instead of disproportionate children with DS since this population also has unusually large tongues, fingers, and other physical features that differ from neurotypical children. The first time the dichotomy is mentioned in line 24, the authors should clearly state that they are referring to head circumference to body weight/mass ratio. Additionally, the authors sometime use the term body weight, other times, body mass. Weight and mass are not technically the same thing.

Another important thing for the authors to make clear is that all children with DS are developmentally delayed, not just those with proportional CI. They often make statements that sound like only the disproportional CI group has delays. Also, when discussing delays in age of milestone acquisition, perhaps use "by" instead of "at" (e.g., delay in prone by 2.81 months).

Below are my suggestions, section by section:

Abstract: In line 29, the word "assess" is used twice in the same sentence; better to use a different word. Lines 32 and 33 should be combined into one sentence. Line 40, change "decisions" to "description". It is also unclear where "a little behind" comes from. Is there a reference for it or do the authors simply mean that clinicians use vague descriptors?

Introduction: Line 49, I recommend changing the wording to something like "greater variability in time range".

Line 53, change to "Two theories have been posited to explain the delay in motor development among children with DS." Conceptually, the focus is on motor development, not on milestones. Milestones are a measurement tool.

Lines 56-57 condense to "...early health issues and various surgeries for associated medical problems..."

Line 59, change "concerns interpopulational differences" to "explains interpopulational differences".

Lines 62-67 is too convoluted. Divide into separate sentences.

Line 71, add "positions" after the phrase "in supine and prone".

Line 99 has a typo. "In" and "different" should be two words. However, it's unclear why the authors listed these references if those studies didn't assess CI. How are they relevant to the current study?

Materials and Methods: Please clarify that the sample was aged 3 months to 5 years at the time of enrollment into the study or baseline.

Line 114, restate that the sample size was not predetermined.

Line 133, change to "CI was defined as HCx100/BW". Also, state who classified the children as proportional/disproportional - the nurses or study authors?

Lines 138-140 are awkwardly worded. It needs to be more clearly explained that the MFDD has NOT been validated on the DS population but you chose that over other instruments that have because it contains more milestones (measurements). The fact that the instruments that were previously used on children with DS who had comorbidities is not relevant to why you didn't use those instruments, it only explains how your study differs from those. Furthermore, it is not clear in the Methods section that you excluded children with comorbidities. I only discovered that in the Discussion section. Please give rationale for excluding those children.

Lines 151-152, seems to contradict lines 123-124. How can HC and BW be measured once a week if some clinical exams occurred every 3-6 months?

Line 155, once again, emphasize MFDD is for the evaluation of motor development, not motor milestones, since you are talking about content validity.

Section 2.4 Outcomes needs revision. The primary outcome is the time to attain different motor milestones. The assessment of reliability of the MFDD is a specific aim, not an outcome measure.

Lines 185-186 is awkwardly worded

Results: How do you know that 9 children were not correctly classified? You need to explain who did the original classification and how you determined that some were misclassified.

Lines 209-213: it is unclear what the numbers in parentheses mean. Typically numbers in parentheses refer to standard deviations, but in this case, the authors have listed the SD as +/- x.xx.

Line 216: it is unclear how Tables 3 and 4 are divided. Earlier versus later motor milestones. The authors should state what the difference is between the two tables and separately state the key points of each table. Currently, the paragraph refers to Tables 3 and 4 and Figure 1 all in one sentence and it is unclear what each table/figure represents. Similarly, it's unclear what the A and B groups mean. As a result, the title of Table 3 is confusing.

All the numbers listed in the tables should go to the same number of digits after the decimal point. Some are of the chi-square values are whole numbers, others go to the tenths place, still others go to the hundredths place. The p-values should either all have a 0 before the decimal point or not.

Line 273, it's not clear what the correlation listed for content validity represents. 

None of the supplementary tables were included in the proof for me to review.

Line 282 is missing a percentage for the people correctly classified in the proportional group.

Table 5, what does the * in the first row signify?

Line 322, clarify that disproportionate children were "more" delayed than proportionate children.

All the numbers listed in the last paragraph of the Results section appear to be reversed. Disproportional CI children should achieve the same milestones at a later age than proportional CI children, not the other way around. Also, it's unclear what variance the authors are referring to. Do they mean that 90% of the children fall within 1 SD of the mean age for that milestone?

Discussion: Reword first sentence to "Our model of motor development in children with DS provides a normative framework for predicting motor milestones in this population."

Lines 343-344 is grammatically incorrect and should be reworded.

Lines 346 should be restated as all children with DS have developmental delay. Do the authors mean when children with DS are even more behind than expected compared to other children with DS?

Lines 366-368 should have also been stated in the Materials and Methods section.

Line 382 contradicts the Introduction section in which the authors stated that children with DS accomplish motor milestones in the same sequence and neurotypical children.

Line 395, unclear what "behind 6.63 times" means. Do you mean more delayed in achieving 6 different milestones compared to proportionate children? Delayed by 6.63 days? Months?

Line 409, end the sentence after data collector. Start new sentence with the word "bias".

Line 418 occurs out of nowhere. There was no mention in the Methods section that the study excluded children with comorbid conditions and why.

Author Response

Response to Reviewer 1 Comments

Point 1: This is a robust longitudinal study that follows the development of 56 children with Down syndrome (DS) over 20 years. That is the strength of the overall study but it is not fully described in this paper as the authors only focus on early motor milestones and do not examine how differences between the proportional and disproportional CI children persist over time or affect long-term development (e.g., speech and language, intelligence, social skills, adaptive behaviors). The paper would be much stronger if they also demonstrate how greater delays in early motor development affect other cognitive domains or persist into middle- or late-childhood or adolescence.

Response 1: We thank the Reviewer for the insightful comment about improving the strength of the present study. It would have been ideal to show additionally how the delay reported in the present study would have affected other cognitive domains or how the reported delays would have affected the children into middle – or late childhood or adolescence. However, the focus of this study was to examine differences in the development of motor milestones and to provide guidance for practitioners’ prediction of the acquisition time of the milestones. Thus, a future study could be designed to determine how differences between proportional and disproportional CI children persist over time or affect long-term development (e.g., speech and language, intelligence, social skills, or adaptive behaviors). We have now added to the last sentence of the Discussion, “Additionally, a future study could determine how differences between children with DS with a proportionate CI and disproportionate CI persist over time or affect long-term development (e.g., speech and language, intelligence, social skills, or adaptive behaviors).”   

Point 2: There are two aims of the study. One, to compare the motor development of children with DS who have proportional head circumference to body mass and those who do not. Two, to test the psychometric properties of the MFDD, which has not been validated for children with DS. The design of the study is able to answer these two aims.

Response 2: Thank you for providing a summary of the aims of the study and a confirmation that we have addressed the aims.

Point 3: There's inconsistent usage of terminology in the abstract. At times, the authors use disproportional/proportional, other times, disproportionate/proportionate. These terms by themselves are somewhat vague as they are adjectives, not nouns. I recommend the authors include "CI" (i.e., disproportionate CI) instead of disproportionate children with DS since this population also has unusually large tongues, fingers, and other physical features that differ from neurotypical children. The first time the dichotomy is mentioned in line 24, the authors should clearly state that they are referring to head circumference to body weight/mass ratio. Additionally, the authors sometime use the term body weight, other times, body mass. Weight and mass are not technically the same thing.

Response 3: We thank the Reviewer for bringing our attention to the inconsistent use of disproportionate/ proportionate and disproportional/proportional throughout the manuscript. We have now accordingly revised the use of the above terms to disproportionate CI, proportionate CI, and so forth throughout the manuscript. We have additionally revised where relevant disproportionate/proportionate or children with DS or disproportional/proportional children with DS to “children with disproportionate/proportionate CI.”

Point 4: Regarding the lack of clarity of what parameters are assessed to determine the cephalation index (CI) in the abstract, we have now added “…the ratio of head circumference and birth weight (HC3/BW) or Cephalization Index (CI)…” so that the definition is clear in the abstract as it is in line 24.

Response 4: We are grateful to the Reviewer for noticing our interchanging use of weight and mass, which we realize are not synonymous. We have now changed “body mass” in the Introduction section to “body weight.” 

Point 5: Another important thing for the authors to make clear is that all children with DS are developmentally delayed, not just those with proportional CI. They often make statements that sound like only the disproportional CI group has delays. Also, when discussing delays in age of milestone acquisition, perhaps use "by" instead of "at" (e.g., delay in prone by 2.81 months).

Response 5: We thank the Reviewer for highlighting that we did not clearly describe that all children with DS are developmentally delayed. We have now added to the second sentence of the Discussion section, “With the knowledge that all children with DS are developmentally delayed, not just children with disproportionate CI, predicting the time to milestone development would enable clinicians to follow the development of motor milestones easily in proportionate and disproportionate children with DS, aged 3 months to 5 years, in three different positions: prone, supine, and standing.” We hope that the revised sentence now clarifies that we did not intend to postulate that children with DS who have proportionate CI do not experience developmental delays.

We have additionally revised the use of “at” to “by” when describing delays in the age of milestone acquisition throughout the manuscript.

Point 6: In line 29, the word "assess" is used twice in the same sentence; better to use a different word. Lines 32 and 33 should be combined into one sentence. Line 40, change "decisions" to "description". It is also unclear where "a little behind" comes from. Is there a reference for it or do the authors simply mean that clinicians use vague descriptors?

Response 6: We are grateful to the Reviewer for the constructive feedback about the sentence structure of the abstract. We have now revised the second occurrence of the word "assess" to "determine" in line 29 of the abstract.

Point 7: Introduction: Line 49, I recommend changing the wording to something like "greater variability in time range".

Response 7: We agree with the helpful comments by the Reviewer and have now revised the wording in line 49 in the sentence to read, "Children with Down syndrome (DS) achieve all motor milestones, such as sitting, crawling, standing, and walking, in the same order as other children, but with a certain delay and greater variability in the time range of motor development compared to typical children."

Point 8: Line 53, change to "Two theories have been posited to explain the delay in motor development among children with DS." Conceptually, the focus is on motor development, not on milestones. Milestones are a measurement tool.

Response 8: We are grateful for the suggested revision to the sentence in line 53. We have now revised the sentence to "Two theories have been posited to explain the delay in motor development among children with DS." We thank the reviewer for the clarification in the distinction between motor development and milestones.

Point 9: Lines 56-57 condense to "...early health issues and various surgeries for associated medical problems..."

Response 9: We have now condensed the sentence to read, "The first is the intrapopulation theory where differences in the acquisition of motor mile-stones in DS children are found in early health issues and various surgeries for associated medical problems which can have a negative impact on motor development."

Point 10: Line 59, change "concerns interpopulational differences" to "explains interpopulational differences".

Response 10: We have now replaced "concerns" with "explains" in line 59.

Point 11: Lines 62-67 is too convoluted. Divide into separate sentences.

Response 11: We thank the Reviewer for the suggestion to divide the long sentence in lines 62-67. We have now separated the previous sentences to read, "Further, previous studies attributed structural and functional disorders of the central nervous system for the delay in the acquisition of motor milestones in children with DS [6, 7]. For example, these disorders include changes in the shape and number of neurons and changes in cerebrum size, hypoplasia and reduced granule cell density of the cerebellum, delayed myelination, and the pathophysiological processes caused by excessive gene expression at chromosome 21 [6,7]."

Point 12: Line 71, add "positions" after the phrase "in supine and prone".

Response 12: We have now added "positions" after the phrase "in the supine and prone".

Point 13: Line 99 has a typo. "In" and "different" should be two words. However, it's unclear why the authors listed these references if those studies didn't assess CI. How are they relevant to the current study?

Response 13: We are thankful for the Reviewers’ suggestions. We have now separated indifferent on line 99 into "in" and "different." We cited the studies by Silventoinen et al. [11] and Sharf et al. [12] because these studies did not assess the CI. Thus, we intended to justify the contribution of our study by providing citations to studies that did not assess the CI in children with DS. We believe the use of the CI is useful in children with DS and our study seeks to address the gap in the literature regarding the utilization of the CI in children with DS. To clarify our rationale for mentioning the references to these studies that did not use the CI, we have now added an additional description to the existing sentence from lines 101 to 102 that reads, "Thus, our study intends to address this gap by seeking to determine…" Now, the complete sentence from lines 102 to 103 reads, "Thus, our study intends to address this gap by seeking to determine whether there is a difference in the acquisition of developmental milestones in children with DS who have a disproportionate CI compared to proportionate CI."

Point 14: Materials and Methods: Please clarify that the sample was aged 3 months to 5 years at the time of enrollment into the study or baseline.

Response 14: We have now added to line 114, “at the time of enrollment or baseline” to the sentence that now reads, “In this prospective longitudinal cohort study, the age of the acquisition of motor development milestones was monitored in 56 children with DS aged 3 months to 5 years at the time of enrollment or baseline and followed up between 2000 and 2020.”

Point 15: Line 114, restate that the sample size was not predetermined.

Response 15: We have now restated in lines 114-115 “The sample size was not predetermined.”

Point 16: Line 133, change to "CI was defined as HCx100/BW". Also, state who classified the children as proportional/disproportional - the nurses or study authors?

Response 16: We agree with the helpful comments by the Reviewer and have now revised the sentence from lines 133 to 134 to “We additionally collected the CI defined as HCx100/BW. “ Additionally, lines 133-134 now state which of the study authors classified the children according to CI, Two authors (ARC and MŠ) additionally classified the children according to the CI defined as HCx100/BW using the cutoff points previously defined by Harel [16] and Simic [17].”

Point 17: Lines 138-140 are awkwardly worded. It needs to be more clearly explained that the MFDD has NOT been validated on the DS population but you chose that over other instruments that have because it contains more milestones (measurements). The fact that the instruments that were previously used on children with DS who had comorbidities is not relevant to why you didn't use those instruments, it only explains how your study differs from those. Furthermore, it is not clear in the Methods section that you excluded children with comorbidities. I only discovered that in the Discussion section. Please give rationale for excluding those children.

Response 17: We are grateful for the Reviewers' constructive comments and have now deleted the text referring to the other studies’ inclusion of children with comorbidities. The sentence from lines 138 to 140 now reads, “We chose the MFDD due the comprehensiveness of the milestones that can be assessed as opposed to other existing instruments [22, 23].” Further, line 121 describes that we excluded children with DS with comorbid conditions; however, we realize that this could be overlooked in this section due to the non-explicit wording that we used. Now, we have revised the wording in the sentence to describe explicitly that we excluded children with comorbidities and a rationale for their exclusion from lines 121 to 123. 

Point 18: Lines 151-152, seems to contradict lines 123-124. How can HC and BW be measured once a week if some clinical exams occurred every 3-6 months?

Response 18: We have now added text to clarify the distinction in the measurement of the children’s’ HC and BW from their clinical examinations in line 151, which now reads “The HC and BW were obtained separately from clinical examinations by specially…” We hope that this now reads clearer than before.

Point 19: Line 155, once again, emphasize MFDD is for the evaluation of motor development, not motor milestones, since you are talking about content validity.

Response 19: We thank the Reviewer for pointing out this oversight. We have now revised this sentence to read, “The reliability and content validity of the MFDD for the evaluation of motor development in children with DS who had proportionate and disproportionate CI was assessed.”

Point 20: Section 2.4 Outcomes needs revision. The primary outcome is the time to attain different motor milestones. The assessment of reliability of the MFDD is a specific aim, not an outcome measure.

Response 20: We have now revised the secondary outcome in section 2.4 Outcomes to “The secondary outcome was the anthropometric parameters including BW, length, and HC for gestational age, and sex of the children with DS.”

Point 21: Lines 185-186 is awkwardly worded

Response 21: We have now revised the sentence from lines 185 to 186 to a sentence that now extends from line 184 to 185, “Our regression model (y =b + mx) attempted to predict the expected months to reach a milestone from the obtained milestone scores, where y as the outcome variable is expected months for a function, b is the intercept, and m is the slope.”

Point 22: Results: How do you know that 9 children were not correctly classified? You need to explain who did the original classification and how you determined that some were misclassified.

Response 22: We have now attached Supplementary Table 4 that show the 9 misclassified children based on the previously published cutoff points by Harel [16] and Simic [17] as the criteria for classification as disproportionate (CI≥1.1) or proportionate (CI<1.1). Similar to Point 16, our response is, “Two authors (ARC and MŠ) additionally classified the children according to the CI defined as HCx100/BW using the cutoff points previously defined by Harel [16] and Simic [17].”  

Point 23: Lines 209-213: it is unclear what the numbers in parentheses mean. Typically numbers in parentheses refer to standard deviations, but in this case, the authors have listed the SD as +/- x.xx.

Response 23: We thank the Reviewer for bringing our attention to the unclear meaning of the numbers in the parentheses. We intended to inform readers about what numbers were used to calculate the average number of examinations. We have now moved the numerators and 56 as the denominator to after the average and standard deviation in lines 209-214. The standard deviation is expressed after the ± sign from lines 209-214.

Point 24: Line 216: it is unclear how Tables 3 and 4 are divided. Earlier versus later motor milestones. The authors should state what the difference is between the two tables and separately state the key points of each table. Currently, the paragraph refers to Tables 3 and 4 and Figure 1 all in one sentence and it is unclear what each table/figure represents. Similarly, it's unclear what the A and B groups mean. As a result, the title of Table 3 is confusing.

Response 24: We appreciate the helpful suggestions that the Reviewer provided regarding Tables 3 and 4 as well as Figure 1. Lines 216-218 now describe Table 3. We have additionally revised the title for Table 3 to highlight that the information shown describes a comparison of early milestones between children with DS who have a proportionate CI and children with a disproportionate CI. We similarly described the purpose of Table 4 and revised its title on the bottom of p. 8. On the top of p. 10, we provided a description of Figure 1 to clarify its purpose. Concerning the reference to groups A and B, we have now provided definitions for walking ages A and B in the footnotes of Table 1 and Figure 1, where the characteristics of each developmental period are described. 

Point 25: All the numbers listed in the tables should go to the same number of digits after the decimal point. Some are of the chi-square values are whole numbers, others go to the tenths place, still others go to the hundredths place. The p-values should either all have a 0 before the decimal point or not.

Response 25: We have now revised the table so that the values other than p-values go the hundredths place. We chose to report the p-values to the thousandths place.

Point 26: Line 273, it's not clear what the correlation listed for content validity represents.

Response 26: The content validity described in line 273 represents how closely the modified Munich Functional Developmental Diagnosis scale represents the original scale in assessing motor development. Specifically, the correlations include the individual milestones with the time of acquisition of walking independently. We have now described this in the text in section 3.3 Validity and reliability of the modified MFDD: “correlations of individual milestones with the time of acquisition of walking independently were 0.930-0.931 [Supplementary Table 1].”

Point 27: None of the supplementary tables were included in the proof for me to review.

Response 27: We thank the Reviewer for bringing our attention to this fact; we have now attached the Supplementary Tables for your review.

Point 28: Line 282 is missing a percentage for the people correctly classified in the proportional group.

Response 28: We have now placed this missing information in lines 282-283.

Point 29: Table 5, what does the * in the first row signify?

Response 29: We have now changed the asterisk to a superscript number 1. We sought to define CI in the footnotes of the table, but simply used the wrong notation.

Point 30: Line 322, clarify that disproportionate children were "more" delayed than proportionate children.

Response 30: We have now placed “more” before disproportionate CI, “Notably, “more disproportionate children were delayed in “crawling age”, “walking age” A and B, and “sitting age” (Tables 3–5).”

Point 31: All the numbers listed in the last paragraph of the Results section appear to be reversed. Disproportional CI children should achieve the same milestones at a later age than proportional CI children, not the other way around. Also, it's unclear what variance the authors are referring to. Do they mean that 90% of the children fall within 1 SD of the mean age for that milestone?

Response 31: We thank the Reviewer for bringing our attention to the reversed order of the reporting of the results in the last paragraph of the Results section. We have now provided the correct descriptions for the corresponding groups of children, as found below:

“Notably, disproportionate children were delayed in “crawling age”, “walking age” A and B, and “sitting age” (Tables 3–5). Children with a proportionate CI acquired the prone position at 2.68 months (p=0.075), accounting for 71% of the variance around the mean months to that milestone. Children with a disproportionate CI acquired the prone position at 3.25 months (p=0.013), accounting for 90% of the variance around the mean months to that milestone. Regarding the supine position, children with a proportionate CI children acquired this milestone at 2.26 months (p=0.011), while those with a disproportionate CI did at 2.39 months (p=0.009). In each of these models, the supine position explained 84 and 85%, respectively, of the variation around the mean months to that milestone. For standing posture A, 4.56 months was the time that children with a disproportionate CI acquired this milestone (p=0.001), whereas the time was 4.60 months for those children with a proportionate CI (p=0.002). We found 94 and 93% variation around the mean time explained by the standing posture A in disproportionate CI and proportionate CI. Children with a disproportionate CI acquired the standing posture B later than proportionate CI (4.50 and 4.09 months, p<0.001 for both), where the standing posture B explained 91% of the variation around the mean months for disproportionate CI. Standing posture B was the variable that explained 91% of the variation around the mean months for children with a proportionate CI.”

The variance means how much the time to achieve a milestone varies around the mean time. Therefore, if there is 90% of variance around the mean, then 90% of children fall within 1 SD of the mean time to reach that milestone. We hope that this explanation provides more information.

Point 32: Discussion: Reword first sentence to "Our model of motor development in children with DS provides a normative framework for predicting motor milestones in this population."

Response 32: We have now replaced the first sentence of the Discussion section with the sentence recommended by the Reviewer.

Point 33: Lines 343-344 is grammatically incorrect and should be reworded.

Response 33: We thank the Reviewer for the suggestion to reword the sentence from lines 343 to 344. We have revised this sentence that includes a revision suggested by Reviewer 1 (point 5 from above), which now reads, '“ With the knowledge that all children with DS are developmentally delayed, not just children with disproportionate CI, predicting the time to milestone development would enable clinicians to follow the development of motor milestones easily in children aged 3 months to 5 years with DS who have proportionate and disproportionate CI in prone, supine, and standing positions.”

Point 34: Lines 346 should be restated as all children with DS have developmental delay. Do the authors mean when children with DS are even more behind than expected compared to other children with DS?

Response 34: We have revised the sentence in line 346 to create two new sentences to clarify that all children with DS have developmental delay and that the model could be used to help those children who are more behind other children with DS.

“The motor development model may allow clinicians to discover when a developmental delay occurs, is it more behind than expected compared to other children with DS. Accordingly, adequate early intervention could be developed to help children with DS with timely physical therapy.”

Point 35: Lines 366-368 should have also been stated in the Materials and Methods section.

Response 35: We have now included an explanation for the reason the MFDD was used in the current study.

Point 36: Line 382 contradicts the Introduction section in which the authors stated that children with DS accomplish motor milestones in the same sequence and neurotypical children.

Response 36: We are grateful to the Reviewer for highlighting the different descriptions about the sequence of motor milestones in children with DS and neurotypical children. We have now inserted text regarding the accomplishment of motor milestones in the two groups of children in the Discussion section in line 382.

“The prevailing opinion in the literature is that children with DS acquire milestones in the same order as healthy children [27, 28]. In our study, children with DS adopted rotations before leaning on extended arms. Leaning on outstretched arms proved to be a more demanding task because it required greater antigravity activity, which was insufficient in the examined children.”

Point 37: Line 395, unclear what "behind 6.63 times" means. Do you mean more delayed in achieving 6 different milestones compared to proportionate children? Delayed by 6.63 days? Months?

Response 37: We thank the Reviewer for pointing out the unclear description about delays. We have now revised the description to describe the value of 6.63 in terms of the likelihood of the acquisition of milestones. The sentence now reads, “In the acquisition thereof, children with a disproportionate CI were 6.63 more likely not to adopt milestones compared to proportionate children.”

Point 38: Line 409, end the sentence after data collector. Start new sentence with the word "bias".

Response 38: We have now corrected this oversight in line 409.

Point 39: Line 418 occurs out of nowhere. There was no mention in the Methods section that the study excluded children with comorbid conditions and why.

Response 39: Similar to our response for point 15, line 121 describes that we excluded children with DS with comorbid conditions; however, we realize that this could be overlooked in this section due to the non-explicit wording that we used. Now, we have revised the wording in the sentence to describe explicitly that we excluded children with comorbidities and a rationale for their exclusion from lines 121 to 123. 

Reviewer 2 Report

Summary: This manuscript is very well-written and the authors should be commended on the quality of research that was conducted. The purpose of the study was to determine the difference in motor milestones among disproportionate and proportionate children with DS using a marker of cephalization index. The secondary purpose was to create a model to predict motor milestones in children with DS according to the proportionality status. It is worth repeating that the manuscript is well written. I have some concerns I would like addressed prior to publication.

Specific comments

The title is very long. A good rule of thumb is ~15 words for a title, and this one is 24. Please shorten.

Key words: Please use key words that are not found in the title of the paper.

Abstract:

Lines 28-30: I believe this sentence is mis-written: “…to assess 28 motor milestones that was assessed for reliability and content validity…” I’m unsure what was assessed for reliability and content validity.

Line 31: insert “and” between acquisition, and controlled for

Lines 39-41: the description of clinicians’ arbitrary decisions that development in children with DS simply lags “a little behind” is not scientifically written. Please revise. Surely clinicians have more standardized terminology that they use.

Introduction

Line 55: Please use person-first language throughout. That is, the term should not be “DS children” but rather “children with DS.” This change should be reflected throughout the entire manuscript.

General comment regarding the introduction: Clearly the crux of this manuscript is the dichotomy of proportionate vs disproportionate children with DS. However, in the (very well-written) introduction, this is explained as the cephalization index (CI) and this term is used more prominently throughout the description of proportionate vs disproportionate children with DS. Thus, it was unclear to me why the terms proportionate and disproportionate were used in the title but CI was not mentioned until the second to last paragraph of the introduction. I would suggest moving this definition up in the introduction section.

Materials and methods:

Lines 121-123: I’m unclear about the exclusion criteria here. What is meant by “during a milestone or group of milestones”?

Line 126: “motor milestone abilities” is redundant. Please remove abilities.

Line 130: Please spell out these acronyms (BW, HC) since this is the first time they are used. If they are not used frequently throughout the remainder of the manuscript, please do not abbreviate them.

In Table 1, please replace all instances of “his or her” with “they” or “their” (whichever is appropriate). This will condense the table and make readability much clearer.

Line 159: I do not think it is necessary to abbreviate discriminant analysis here or in the remainder of the manuscript. I find it a but confusing with the more commonly used DS acronym used throughout.

Line 182: A quotation mark is missing after “walking age A and B”

The statistical analysis section is very well written and easy to follow.

Figure 1: The individuals regression models do not appear to be labeled as A, B, C, D but they are explained in such a way in the description of the figure.

Lines 258-265: Please just use “CI” throughout this description rather than spelling out cephalization index.

Lines 280-284: I think this is a run-on sentence. Please rewrite.

Line 135: Should this be ‘proportionate’ rather than ‘proportional’?

Line 137: Same comment here and throughout the rest of the paragraph. “Proportionate” and “disproportionate” have been used consistently throughout the manuscript up to this point, but suddenly “proportional” and “disproportional” are used here.

Line 349: Please insert the word “children” after “proportionate”

Line 379: person-first language, please

Line 395: Reword to “…children were 6.63 times behind compared to…”

Line 424: Can you please provide an example of what a third type of body disproportionality would be?

Author Response

Response to Reviewer 2 Comments

Point 1: Summary: This manuscript is very well-written and the authors should be commended on the quality of research that was conducted. The purpose of the study was to determine the difference in motor milestones among disproportionate and proportionate children with DS using a marker of cephalization index. The secondary purpose was to create a model to predict motor milestones in children with DS according to the proportionality status. It is worth repeating that the manuscript is well written. I have some concerns I would like addressed prior to publication.

Response 1: We are grateful to the Reviewer for the constructive comments on the present research study, particularly regarding the robustness of the writing.

Specific comments

Point 2: The title is very long. A good rule of thumb is ~15 words for a title, and this one is 24. Please shorten.

Response 2: We appreciate the suggestion to shorten the title. We have now shortened the title to read, “Acquisition of motor milestones in proportionate and disproportionate children with Down syndrome: a prospective cohort study.”

Point 3: Key words: Please use key words that are not found in the title of the paper.

Response 3: We have now replaced the keywords that appear in the title of the paper with words not  found in the title. The new keywords are now longitudinal design and gross motor development.

Abstract:

Point 4: Lines 28-30: I believe this sentence is mis-written: “…to assess 28 motor milestones that was assessed for reliability and content validity…” I’m unsure what was assessed for reliability and content validity.

Response 4: We are thankful to the Reviewer 2 for pointing out this sentence for which Reviewer 1 also suggested a revision. We have now revised the sentence from lines 28-30 to be clearer. The sentence now reads, “We used a modified Munich Functional Developmental Diagnostic Scale (MFDD) that was assessed for reliability and content validity (Levene's test and discriminant analysis) to determine 28 motor milestones.” 

Point 5: Line 31: insert “and” between acquisition, and controlled for

Response 5: We have inserted “and” between “acquisition, and controlled for” on line 31 in the abstract.

Point 6: Lines 39-41: the description of clinicians’ arbitrary decisions that development in children with DS simply lags “a little behind” is not scientifically written. Please revise. Surely clinicians have more standardized terminology that they use.

Response 6: We appreciate your suggestion to clarify the meaning of "a little behind" in the abstract. This suggestion is the same as the first Reviewer's suggestion in Point 6. We realize that the text that reads "a little behind" can confuse readers regarding its origin. Thus, we have now replaced this with a description of the timing of developmental delays as, "about the timing of developmental delays in children with DS" in line 40.

Introduction

Point 7: Line 55: Please use person-first language throughout. That is, the term should not be “DS children” but rather “children with DS.” This change should be reflected throughout the entire manuscript.

Response 7: We have now replaced all occurrences of “DS children” with “children with DS” throughout the manuscript.

Point 8: General comment regarding the introduction: Clearly the crux of this manuscript is the dichotomy of proportionate vs disproportionate children with DS. However, in the (very well-written) introduction, this is explained as the cephalization index (CI) and this term is used more prominently throughout the description of proportionate vs disproportionate children with DS. Thus, it was unclear to me why the terms proportionate and disproportionate were used in the title but CI was not mentioned until the second to last paragraph of the introduction. I would suggest moving this definition up in the introduction section.

Response 8: We thank the Reviewer for suggesting that we move the definition of the CI up earlier in the Introduction section. However, the preceding paragraphs to the second to last paragraph in the Introduction section are intended to familiarize readers with what is already known to justify the current study. We placed the definition of the CI later to prepare readers for our objectives.

Materials and methods:

Point 9: Lines 121-123: I’m unclear about the exclusion criteria here. What is meant by “during a milestone or group of milestones”?

Response 9: We realize that the wording may confuse readers. We have now revised “during” to “by the time of a” to describe when a child has acquired a certain milestone or a group of milestones.

Point 10: Line 126: “motor milestone abilities” is redundant. Please remove abilities.

Response 10: We have now removed the word “abilities” from line 126.

Point 11: Line 130: Please spell out these acronyms (BW, HC) since this is the first time they are used. If they are not used frequently throughout the remainder of the manuscript, please do not abbreviate them.

Response 11: We have now defined the abbreviations for BW and HC in line 130, since these abbreviations are used throughout the manuscript.

Point 12: In Table 1, please replace all instances of “his or her” with “they” or “their” (whichever is appropriate). This will condense the table and make readability much clearer.

Response 12: We have now replaced all instances of “his or her” with “themselves” or “their” in Table 1.

Point 13: Line 159: I do not think it is necessary to abbreviate discriminant analysis here or in the remainder of the manuscript. I find it a but confusing with the more commonly used DS acronym used throughout.

Response 14: We are grateful to the Reviewer for highlighting the use of the abbreviation for discriminant analysis that could be confused with the abbreviation for Down syndrome. We have now spelled out discriminant analysis and deleted the abbreviation for it throughout the manuscript.

Point 15: Line 182: A quotation mark is missing after “walking age A and B”

Response 15: We have now placed the closing quotation marks to so that line 182 contains (“walking age A and B”).

Point 16: The statistical analysis section is very well written and easy to follow.

Response 16: We appreciate the constructive feedback about the writing and reading ease of the statistical analysis section.

Point 17: Figure 1: The individuals regression models do not appear to be labeled as A, B, C, D but they are explained in such a way in the description of the figure.

Response 17: We have now labeled the individual panels of the regression models in Figure 1 to reflect the labeling that is in the legend.

Point 18: Lines 258-265: Please just use “CI” throughout this description rather than spelling out cephalization index.

Response 18: We have now used the abbreviation CI throughout the caption for Figure 1.

Point 19: Lines 280-284: I think this is a run-on sentence. Please rewrite.

Response 19: Line 282 was missing a percentage for the people correctly classified in the proportional group, which we have now inserted to complete the sentence. Prior to this, the missing information made the sentence hard to understand.

Point 20: Line 135: Should this be ‘proportionate’ rather than ‘proportional’?

Response 20: In response to Reviewer 1 and your comments about the use of disproportional/proportional, we have changed all occurrences to disproportionate/proportionate throughout the manuscript.

Line 137: Same comment here and throughout the rest of the paragraph. “Proportionate” and “disproportionate” have been used consistently throughout the manuscript up to this point, but suddenly “proportional” and “disproportional” are used here.

Response 21: This comment is the same as above in response to Reviewer 1 and your comments about the use of disproportional/proportional, we have changed all occurrences to disproportionate/proportionate throughout the manuscript.

Point 22: Line 349: Please insert the word “children” after “proportionate”

Response 22: We have now inserted the word “children” after “proportionate” in line 349.

Point 23: Line 379: person-first language, please

Response 23: We have now revised the sentence starting on line 379 with “For instance, children with DS in our study acquired “Rolls both ways” before they acquired “Extended arm support” and “Sits alone stably” before “Sits down alone”.” We inserted “our” before “study.” 

Point 24: Line 395: Reword to “…children were 6.63 times behind compared to…”

Response 24: We thank the Reviewer for pointing out the unclear description about delays that Reviewer 1 also suggested to revise. We have now revised the description to describe the value of 6.63 in terms of the likelihood of the acquisition of milestones. The sentence now reads, “In the acquisition thereof, children with a disproportionate CI were 6.63 more likely not to adopt milestones compared to proportionate children.”

Point 25: Line 424: Can you please provide an example of what a third type of body disproportionality would be?

Response 25: We thank the Reviewer for bringing our attention to a description that we intended to describe as “other forms of disproportionality”. We used the CI (HC3/BW) as a measure of dis/proportionality. Disproportionality can also be the result of discrepancies in other anthropometric measures, e.g., body weight and body length, ponderal index, PT/PD, or BMI. We have now inserted a reference to a study that reports about other determinants of dis/proportionality in children in the last paragraph of the Discussion section.

Round 2

Reviewer 1 Report

There is no need to repeatedly state in the Results section that the sample is of children with DS since there are no children without DS in the study. Focus the discussion on proportionate vs. disproportionate CI.

TITLE: Add the term cephalization index to the title. I suggest something like "Acquisition of motor milestones in children with Down syndrome based upon proportionality of cephalization index: a prospective cohort study".

ABSTRACT: Write out what CI stands for the first time it is mentioned (line 26). Line 35, delete "and" change to "controlling for for sex, maternal age..."

INTRODUCTION

1. Lines 56-57, omit "of motor development". Line 58, omit "they". Line 59, omit "of" before the number of months.

2. Line 63, change "are found in [early health issues]" to "believed to be caused by" or "attributed to".

3. Line 69, delete "that cause delays in motor milestone acquisition". It is redundant since the entire paragraph is about that.

4. Lines 72-73, change to [disorders of the central nervous system] "can delay motor development."

5. Lines 97-98 have awkward wording. Better to say the index was reliable and why rather than personifying the word index.

6. Lines 99-100 are unclear. Cannot tell how Harel and Simic calculated proportionality differently than Nishi et al.

7. Lines 104-105, change to "delay in establishing head control, a fundamental motor task..."

8. Lines 111-112, change to "unknown whether CI of children with DS predict future gross motor development".

9. Line 115, add "those with" before proportionate CI.

MATERIALS AND METHODS:

1. State which specific comorbid conditions were excluded. Children with DS are known to have numerous comorbidities including congenital heart defects. 

2. Instead of using the term "children with DS" over and over again, just use "participants". Some sentences, such as line148 don't need any reference to children. It's clear that the nurses were collecting weight, head circumference, etc. of the participants.

3. Section 2.5, verbs should consistently be in the past tense. Also, rather than constantly saying disproportionate vs. proportionate, you can say compared the two groups. It is unclear if all 28 milestones were used as predictor variables or a total score. 

4. Line 200, delete "for the motor developmental milestones".

5. Reference to Supplementary Table 1 in text is missing.

RESULTS:

1. Line 220, change to "...9 (16%) children were excluded due to misclassification of CI proportionality..."

2. Lines 239-241, change "### procedures out of 56" to "for 56".

3. Line 247, change "compared to" to "earlier than".

4. Table 3 title, change to "Comparison of early motor milestone acquisition between children with DS with proportionate and disproportionate CI".

5. Lines 267-268, move the word milestones: "...children with a proportionate CI acquired all milestones, except..., earlier".

6. Figure 1 caption, change acquiring to "acquisition".

7. Line 323, change to "The modified MFDD had good psychometric properties, as indicated by its high reliability..."

8. Supplementary table 3 was not provided. Unclear if the authors forgot to include the submission or if S. Tables 4 and 5 should be renumbered.

9. Lines 337-338 should be worded in the same format as lines 339-340, which is more concise.

10. Table 5 title should be changed to "...between proportionate and disproportionate CI".

11. Line 388, change to "more delayed...than proportionate children".

12. Last paragraph of Results section reads as though the authors used the acquisition time of an earlier milestone to explain the variation in mean months of a later milestone. If so, I suggest summarizing this method at the beginning of the paragraph. Also, the current wording sounds circular - the supine position explains the variation to that milestone. Did the authors mean that proportionality of CI explained the variation in mean months to the supine position?

DISCUSSION:

1. Change wording in 1st paragraph to "...the ability to predict the time to prone, supine, and standing positions would enable clinicians to better track motor development in children..."

2. Lines 418-420, change to "Our model of motor development by proportionality of CI allows clinicians to identify when an unexpected delay occurs..."

3. Line 440, Omit all the verbiage before the word "although". Start the sentence there.

4. Line 446, change to "whereby the proportionality of the children could explain differences in motor development".

5. Link the findings back to the theories presented in the Introduction. It appears that this study supports studies that suggest physical conditions (e.g., muscle tone, hypermobility) explains more of the delay in motor milestone acquisition than health comorbidities and CNS irregularities.

6. Lines 452-453, give examples of differences found in the study cited.

7. Lines 473-474, is this order different in typical children?

8. Lines 490, change "data collector" to rater.

9. Limitations section, add a summary sentence such as, these likely minimized differences between raters.

10. Line 506, include the fact that misclassification was determined through discriminant analysis.

11. Lines 511-513, calculating dis/proportionality based upon Apgar scores or respiratory disorders seems very different than using BW and HC like in the current study.

12. Line 519, change to "...affect development in other domains".

Author Response

10.12.2022

Response to Reviewer 1 Comments

Comments and Suggestions for Authors

Point 1: There is no need to repeatedly state in the Results section that the sample is of children with DS since there are no children without DS in the study. Focus the discussion on proportionate vs. disproportionate CI.

Response 1: We thank the Reviewer for pointing out the repetition. We have now removed repetitive references to “children with DS” in the Results section, with the exception of table titles. We have deleted references to “with DS” from line 220 and “children with DS” from 242.

Point 2: TITLE: Add the term cephalization index to the title. I suggest something like "Acquisition of motor milestones in children with Down syndrome based upon proportionality of cephalization index: a prospective cohort study".

Response 2: We have now revised the title to include cephalization index. The revised title now reads, “Impact of neonatal body (dis)proportionality determined by the Cephalization Index (CI) on gross motor development in children with Down syndrome: A prospective cohort study.”

Point 3: ABSTRACT: Write out what CI stands for the first time it is mentioned (line 26). Line 35, delete "and" change to "controlling for for sex, maternal age..."

Response 3: We have now defined cephalization index when we first mentioned it in line 26. Additionally, we have deleted “and” in line 35 and changed “controlled” to “controlling” in line 35.

INTRODUCTION: 

Point 4: Lines 56-57, omit "of motor development". Line 58, omit "they". Line 59, omit "of" before the number of months.

Response 4: We are grateful to the Reviewer for the suggested edits. We have now omitted "of motor development" from lines 56-57. In line 58, we have now omitted "they" and in line 59, we omitted "of" before the number of months.

Point 5: Line 63, change "are found in [early health issues]" to "believed to be caused by" or "attributed to".

Response 5: We have now changed "are found in [early health issues]" to "attributed to" in line 63.

Point 6: Line 69, delete "that cause delays in motor milestone acquisition". It is redundant since the entire paragraph is about that.

Response 6: We thank the Reviewer for the suggestion. We have now deleted “that cause delays in motor milestone acquisition" from line 69.

Point 7: Lines 72-73, change to [disorders of the central nervous system] "can delay motor development."

Response 7: We have now changed the sentence to read, “Further, previous studies showed that structural and functional disorders of the central nervous system can delay motor development [6,7].”

Point 8: Lines 97-98 have awkward wording. Better to say the index was reliable and why rather than personifying the word index.

Response 8: We have now revised this sentence to read that the index was reliable in lines 97-98 rather than personifying the index.

Point 9: Lines 99-100 are unclear. Cannot tell how Harel and Simic calculated proportionality differently than Nishi et al.

Response 9: We have now revised line 99-100 to read, “Meanwhile, Harel [16] and Simic [17] similarly calculated neonate proportionality by looking at the HC and body weight and further described the relationship as the cephalization index (CI).” In this way, we hope to clarify that Harel and Simic described the relationship between HC and BW as the CI, as opposed to Nishi et al.

Point 10: Lines 104-105, change to "delay in establishing head control, a fundamental motor task..."

Response 10: We have now changed lines 104-105 to “delay in establishing head control, a fundamental motor task...”

Point 11: Lines 111-112, change to "unknown whether CI of children with DS predict future gross motor development".

Response 11: We thank the Reviewer for the suggestion. We have now changed lines 111-112 to “It is unknown whether the CI of children with DS predicts future gross motor development.”

Point 12: Line 115, add "those with" before proportionate CI.

Response 12: We have now added add "those with" before proportionate CI in line 115.

MATERIALS AND METHODS:

Point 13: State which specific comorbid conditions were excluded. Children with DS are known to have numerous comorbidities including congenital heart defects. 

Response 13: We have now described the comorbid conditions that were excluded from the study in the second paragraph of the Materials and Methods section.

Point 14: Instead of using the term "children with DS" over and over again, just use "participants". Some sentences, such as line148 don't need any reference to children. It's clear that the nurses were collecting weight, head circumference, etc. of the participants.

Response 14: We have now replaced several instances of the repetition of “children with DS” with “participants” on line 148 and throughout the Materials and Methods section where relevant.

Point 15: Section 2.5, verbs should consistently be in the past tense. Also, rather than constantly saying disproportionate vs. proportionate, you can say compared the two groups. It is unclear if all 28 milestones were used as predictor variables or a total score. 

Response 15: We thank the Reviewer for pointing out the inconsistent use of verbs in the past tense. On line 197, we changed differ to “differed.” Concerning the constant use of disproportionate vs. proportionate, we have now added “between the two groups” from lines 197 to 198. We used the present tense to describe the regression model equation, as is customary. Additionally, we have added “28” before developmental milestones to clarify that all 28 milestones were used as predictor variables on line 201.

Point 16: Line 200, delete "for the motor developmental milestones".

Response 16: We have deleted "for the motor developmental milestones" from line 200.

Point 17: Reference to Supplementary Table 1 in text is missing.

Response 17: We have now referred to Supplementary Table 1 in section 3.3.

RESULTS:

Point 18: 1. Line 220, change to "...9 (16%) children were excluded due to misclassification of CI proportionality..."

Response 18: We have now changed the sentence on line 220 to “…9 (16%) children with DS that were excluded due to misclassification of the CI not classified according to…”

Point 19: Lines 239-241, change "### procedures out of 56" to "for 56".

Response 19: We have now changed "### procedures out of 56" to "for 56" from lines 239 to 241.

Point 20: Line 247, change "compared to" to "earlier than".

Response 20: We have now changed "compared to" to "earlier than" on line 247.

Point 21: Table 3 title, change to "Comparison of early motor milestone acquisition between children with DS with proportionate and disproportionate CI".

Response 21: We thank the Reviewer for the suggestion. We have now revised the title of Table 3.

Point 22: Lines 267-268, move the word milestones: "...children with a proportionate CI acquired all milestones, except..., earlier".

Response 22: We have now moved the word milestones after “all” in the sentence.

Point 23: Figure 1 caption, change acquiring to "acquisition".

Response 23: We have now changed acquiring to "acquisition".

Point 24: Line 323, change to "The modified MFDD had good psychometric properties, as indicated by its high reliability..."

Response 24: We have now revised the first sentence in section 3.3 on line 323 to read, “The modified MFDD had good psychometric properties, as indicated by the psychometric features of the modified MFDD determined its high reliability (Cronbach’s alpha= 0.933) and content validity (0.930–0.931).”

Point 25: Supplementary table 3 was not provided. Unclear if the authors forgot to include the submission or if S. Tables 4 and 5 should be renumbered.

Response 25: We are grateful to the Reviewer for noticing that Supplementary Table 3 was not included, which was unintended. We have now included it along with the other Supplementary Tables ordered the same way in the manuscript.

Point 26: Lines 337-338 should be worded in the same format as lines 339-340, which is more concise.

Response 26: We have now re-worded lines 337-338, which now reads,”… 86.7% (24/27) of them were properly classified in the proportionate group, and 13.3% (4/30) were wrongly classified.”

Point 27: Table 5 title should be changed to "...between proportionate and disproportionate CI".

Response 27: We have now changed the title of Table 5 to “Comparison of the acquisition of sitting and protective extension reflex motor skills between proportionate and disproportionate CI.”

Point 28: Line 388, change to "more delayed...than proportionate children".

Response 28: We have now added, "more delayed...than proportionate children" to the sentence on line 388.

Point 29: Last paragraph of Results section reads as though the authors used the acquisition time of an earlier milestone to explain the variation in mean months of a later milestone. If so, I suggest summarizing this method at the beginning of the paragraph. Also, the current wording sounds circular - the supine position explains the variation to that milestone. Did the authors mean that proportionality of CI explained the variation in mean months to the supine position?

Response 29: We thank the Reviewer for the insightful comments. We have now provided a summary of the method of using the acquisition time of an earlier milestone to explain the variation in mean months of a later milestone in the first sentence of the last paragraph of the Results section. Additionally, we have now revised the wording surrounding the supine position to “In each of these models, the proportionality of the CI explained 84 and 85%, respectively, of the variation around the mean months to the supine position.”

DISCUSSION:

Point 30: Change wording in 1st paragraph to "...the ability to predict the time to prone, supine, and standing positions would enable clinicians to better track motor development in children..."

Response 30: We have now revised the sentence referred to by the Reviewer in the Discussion section. The sentence now reads, “With the knowledge that all children with DS are developmentally delayed, not just children with disproportionate CI, the ability to predict the time to prone, supine, and standing positions would enable clinicians to better track motor development in children aged 3 months to 5 years with DS who have proportionate and disproportionate CI.”

Point 31: Lines 418-420, change to "Our model of motor development by proportionality of CI allows clinicians to identify when an unexpected delay occurs..."

Response 31: We have now revised the sentence on lines 418-420 to read, “Our model of motor development by the proportionality of the CI allows clinicians to identify when an unexpected delay occurs…”

Point 32: Line 440, Omit all the verbiage before the word "although". Start the sentence there.

Response 32: We have now revised the sentence to begin with “although.”

Point 33: Line 446, change to "whereby the proportionality of the children could explain differences in motor development".

Response 33: We have now changed the sentence on line 446 to read, “Additionally, in contrast to the Lauteslager [26] study, we compared the time to reach motor milestones between children with DS who have disproportionate and proportionate CI, whereby the proportionality of the children could explain differences in motor development.”

Point 34: Link the findings back to the theories presented in the Introduction. It appears that this study supports studies that suggest physical conditions (e.g., muscle tone, hypermobility) explains more of the delay in motor milestone acquisition than health comorbidities and CNS irregularities.

Response 34: We have now provided text to link our findings back to the theories presented in the Introduction section. The last sentence of the fourth paragraph in the Discussion section reads, “Our findings support interpopulation interpretations from previous studies that suggested that the CI physical conditions are instrumental in the delay in motor milestone acquisition rather than chronic diseases or CNS irregularities [2,4,6,7].

Point 35: Lines 452-453, give examples of differences found in the study cited.

Response 35: We have now provided an example of children with DS acquiring the prone position later than typical children as found by Tudella et al. We have accordingly revised the citation to 28 rather than 23. We have now provided a new sentence that reads, “Tudella and colleagues [28] found that children with DS acquired the prone position in their 8th and 9th months compared to typical children who usually acquire prone in the 6th month.”

Point 36: Lines 473-474, is this order different in typical children?

Response 36: We thank the Reviewer for the helpful comment. We have now clarified whether jumping in place is acquired in a different order than typical children by adding to the existing sentence on lines 473-474. The sentence now reads, “The most complex and difficult motor function to evaluate was jumping in place, which develops last in children with DS just as in typical children except with delays [28].”

Point 37: Lines 490, change "data collector" to rater.

Response 37: We have now changed "data collector" to rater on line 490.

Point 38: Limitations section, add a summary sentence such as, these likely minimized differences between raters.

Response 38: We have now added “…which likely minimized differences between the raters” to the sentence that now reads, “However, clinicians involved in the data collection followed standard protocols for the procedures, which likely minimized differences between the raters.”

Point 39: Line 506, include the fact that misclassification was determined through discriminant analysis.

Response 39: We have now added “…according to discriminant analysis” to line 506.

Point 40: Lines 511-513, calculating dis/proportionality based upon Apgar scores or respiratory disorders seems very different than using BW and HC like in the current study.

Response 40: We have now deleted the text from 507 to 514 and the associated reference due to the difference between the current study and the study that used parameters such as Apgar scores and respiratory disorders to calculate dis/proportionality in the paragraph after the Limitations section.

Point 41: Line 519, change to "...affect development in other domains".

Response 41: We have now changed the end of the sentence on line 519 to "...affect development in other domains".
